# Prediction of Overall Survival by Thymidine Kinase 1 Combined with Prostate-Specific Antigen in Men with Prostate Cancer

**DOI:** 10.3390/ijms24065160

**Published:** 2023-03-08

**Authors:** Bernhard Tribukait, Per-Olof Lundgren, Anders Kjellman, Ulf Norming, Claes R. Nyman, Kiran Jagarlmundi, Ove Gustafsson

**Affiliations:** 1Department of Oncology-Pathology, Karolinska Institute and University Hospital Solna, 141 86 Stockholm, Sweden; 2Cancer Centrum Karolinska, CCK, Plan 00, Visionsgatan 56, Karolinska Universitetssjukhuset, Solna, 171 64 Stockholm, Sweden; 3Department of Clinical Science, Intervention and Technology, Karolinska Institute and Karolinska University Hospital, 141 86 Stockholm, Sweden; 4Department of Clinical Science and Education, Södersjukhuset, 118 83 Stockholm, Sweden; 5Research and Development Division, AroCell AB, 111 52 Stockholm, Sweden

**Keywords:** biomarker integration, PSA, thymidine kinase 1, prostate cancer, prediction, survival

## Abstract

Thymidine kinase 1 (TK1) is an intracellular enzyme involved in DNA-precursor synthesis. Increased serum TK1 levels are used as a biomarker in various malignancies. We combined serum TK1 with PSA and evaluated its capacity to predict overall survival (OS) in 175 men with prostate cancer (PCa), detected by screening in 1988–1989 (*n* = 52) and during follow-up (median 22.6 years) (*n* = 123). TK1 was measured in frozen serum, age was stratified into four groups, and dates of PCa diagnosis and dates of death were obtained from Swedish population-based registries. The median concentration of TK1 and PSA was 0.25 and 3.8 ng/ml. TK1 was an independent variable of OS. In the multivariate analysis, PSA was not statistically significant in combination with age whereas the significance remained for TK1 + PSA. Measured once, TK1 + PSA predicted a difference of up to 10 years (depending on patient subgroup) in OS at a median of 9 years before PCa diagnosis. The TK1 concentration in 193 controls without malignancies did not differ from that of the PCa patients, hence TK1 was likely not released from incidental PCa. Thus, TK1 in the blood circulation may indicate the release of TK1 from sources other than cancers, nonetheless associated with OS.

## 1. Introduction

Prostate-specific antigen (PSA) is one of the most used blood biomarkers. In individual patients with newly detected or recurrent prostate cancer (PCa) and for disease monitoring, PSA is a cornerstone of risk stratification [1,2]. PSA is also used for the screening of PCa [2,3,4]. The screening of asymptotic men is motivated by the high incidence and high rate of death from PCa. Worldwide, PCa is one of the most frequently diagnosed malignancies in men. In Western Europe and North America, incidence rates are about 75 per 100,000 males, of whom 10–15% will die from the disease [5,6]. The rising incidence of PCa has partly been attributed to the extensive use of PSA measurements [7,8]. The specificity of PSA for PCa is relatively low. For instance, in the ERSPC (European Randomized Study for Screening of Prostate Cancer) PSA-guided trial, >20,000 biopsies were performed in about 72,000 men during 16-year follow-up with a positive predictive value for PSA of 24%. More than half of the detected tumors were of low risk (T1), likely not fatal for the patient [9].

The aim of several novel tissue-, blood- and urine-based biomarkers, (including sub forms of PSA, enzymes other than PSA or genetic and epigenetic signatures) is to increase the diagnostic, prognostic, and predictive capability [10,11]. For several reasons, molecular markers are not used in routine pathology, e.g., insufficient evidence, lack of standardization, complexity of the methods and cost of commercially available biomarkers [12]. In the screening setting, when efforts are made to detect clinically significant cancers only, biomarkers have been combined with clinical information, e.g., in the Stockholm-3 risk model, various molecular forms of PSA are combined with markers of genetic predisposition and family history [13,14].

Blood-borne molecules are often defined as markers of malignant transformation when concentrations exceed a level presumed to be normal. For instance, the serum concentration of thymidine kinase 1 (TK1) in normal individuals was found to be 0.25 ng/ml [15]. Tk1 is a key salvage enzyme in DNA synthesis localized in the cytosol of all proliferating cells. Thymidine, reutilized from degraded DNA, diffuses from the extracellular space into the cell and is catalyzed by TK1, converted to thymidine monophosphate, and used in the synthesis and repair of DNA (Figure 1) [16,17].

The concentration of TK1 is cell cycle related, mimicking the rate of DNA synthesis: it is unmeasurable in the G0/G1 phase, increases at the G1/S-phase border, peaking in S-phase/G2 and is finally degraded at mitosis by ubiquitination [18]. An isoenzyme (TK2) is found in mitochondria but is unrelated to the cell cycle [16]. Increased blood concentrations of TK1 are found in patients with hematologic malignancies and in patients with various solid tumors [19,20]. Possibly, TK1 is released from proliferating, though non-viable tumor cells. Thus, increased blood concentrations of TK1 might reflect tumor cell loss [21], which is an important factor of tumor growth [22]. The source and biological significance of 0.2–0.3 ng/ml of serum TK1 found in a normal, seemingly healthy population is unknown [15].

A limited number of studies address serum TK1 measurements and PCa [23,24,25,26,27,28]. Despite differences in methodology (assessment of enzyme activity or protein content), heterogeneity of the tumors (localized, metastatic, etc.), different treatments and study aims and endpoints, some conclusions can be drawn: 1. high TK1 values at diagnosis of PCa indicate existence of distant metastases; 2. TK1 seems to be independent of PSA and, therefore, the combination of both markers could improve the diagnostic and prognostic strength in the evaluation of PCa.

In a previous investigation, we studied whether the blood concentration of TK1 at baseline was associated with future PCa-specific mortality [28]. Frozen serum samples from 1988–1989 were thawed and analyzed. After 30 years of follow-up, the time from diagnosis to death differed by 10 years in men with serum TK1 levels above and below the median. This demonstrates the feasibility to obtain prognostic information by analyzing TK1 in frozen serum, after 30 years.

In this study, we evaluated whether TK1, combined with PSA, could provide prognostic information on future death in men aged 55–70 years with PCa, either diagnosed at screening or during 30 years of follow-up. The endpoint was overall survival (OS). Since age is an obvious a potential confounder when evaluating OS, and, also, is associated with PSA, OS was evaluated for age-stratified groups.

## 2. Results

### 2.1. Descriptive Evaluation of the Study Cohort and Controls

Between 1988 and 1989, a group of 1782 men with a median age of 63.7 (range 54.5–71.5) years underwent screening for PCa. Median follow-up time was 18.6 years. By the end of 2018, i.e., after 30 years of follow-up, 299 men (16.8%) were alive. PCa was diagnosed in 271 men (15.2%), 65 at screening (±1 year) and the remaining after a median follow-up of 14.4 years. Baseline characteristics and follow-up-related findings for the PCa group and the 271 controls are listed in Table 1. Although the groups were age matched, they differed not only with respect to prostate-related properties. The men with PCa reached a significantly higher age and a higher proportion of men was alive at last follow-up; other malignancies (squamous cell carcinoma of the skin were excluded) were twice as many in the control group. Although these men were diagnosed at the same age as the men diagnosed with PCa, survival of controls with other malignancies was significantly shorter. The four most frequent malignancies in the 48 men with PCa were colon/rectum 29%, malignant melanoma 19%, urinary bladder 17% and lung/pancreas 10%, and in the 78 controls: lung/pancreas 24%, colon/rectum 15%, urinary bladder 15% and esophagus/liver 9%. There was no statistically significant difference of TK1 concentration in men with PCa and in the controls.

Figure 2 shows the interval from screening to diagnosis and survival after diagnosis among the 271 men with PCa. For the subgroup diagnosed at screening, the median time to last follow-up was 12.4 years. Among those who were diagnosed during follow-up, the corresponding values were 6.8 years. The four dots on the diagonal line represent cases with PCa diagnosed at autopsy.

Changes in age composition of the PCa group during follow-up are shown in Appendix A. The median age at last follow-up of the men with screening-detected PCa was 77 years and less than 10% were alive whereas men with interval-detected PCa were diagnosed at the age of 84 years and the median age was 88 years at last follow-up. Nearly half of them were alive. Regarding prostate-related properties, there was no significant relationship between prostate volume at screening and the time interval between screening and diagnosis of PCa. However, there was a significant relationship between PSA and time from screening to diagnosis (*p* < 0.0001), for free/total PSA ratio (*p* < 0.0001) and for PSA density (*p* < 0.0001). Notably, there was no relationship between TK1 at screening and time to diagnosis. Other malignancies, about 18%, were equally distributed over the different time periods.

### 2.2. Survival in Relation to PSA, TK1 and TK1 + PSA

TK1 concentration was available for 175 of the 271 men with PCa. As shown in Appendix A, most other screening variables did not differ between the 175 men with TK1 values and the 96 men without TK1 data. The median PSA and PSA densities in the former group (3.8 ng/ml and 0.14) were, however, somewhat higher than in the latter one (2.6 ng/ml and 0.13, respectively).

The cohort was dichotomized based on median concentration of PSA and TK1 for further survival analysis. The median survival (95% CI) of the two PSA groups was 23.4 (19.6–25.8) years and 16.8 (13.8–19.3) years (*p* = 0.0017) (Figure 3a). The corresponding values for the TK1 groups were 22.6 (19.6–24.4) years and 17.8 (14.8–19.9) years (*p* = 0.0177) (Figure 3b).

Combining the two dichotomized markers yields four subgroups: (1) LowTK1 + LowPSA, (2) LowTK1 + HighPSA, (3) HighTK1 + LowPSA, (4) HighTK1 + HighPSA. The median survival of the subgroups was 24.8 (20.2–27.6), 19.7 (14.3–22.5), 21.5 (17.6–25.8) and 14.3 (12.2–18.0) years (*p* = 0.003) (Figure 3c). The prognostic value of PSA and TK1 in combination is higher than that for the individual markers. The difference in OS is 10.5 years between group 1 and group 4 compared to 6.6 years for the PSA strata alone. The Kaplan–Meier estimates do not allow the adjusting for age. In the two TK1 groups, mean age was 63.3 and 63.3 years, whereas the age of the PSA groups differed (62.2 and 64.4 years *p* = 0.0002).

### 2.3. Survival in Relation to TK1 and Age

The 175 men with PCa were sub-grouped according to age into four quartiles with medians (IQR) of 58.2 (2.6), 62 (1.4), 64.8 (1.6) and 68.2 (1.4) years. The median survival time (95% CI) was 28.0 (23.7– ), 22.2 (18.5–26.3), 18.8 (14.4–22.2) and 13.4 (10.9–16.7) years (Appendix A), and survival probability (SP) was 32.6 (28.4–40.0), 26.5 (23.0–31.3), 20.2 (17.7–22.9) and 15.5 (13.6–17.6) years. The difference in age between quartile 1 and 4 was 10 years, but median survival differed by about 15 years and SP by about 17 years.

The SP of the four age groups with a TK1 concentration <0.25 ng/ml decreased step by step with age (Figure 4a), and death started to occur after >10 years, 6 years and within 2 years after screening illustrated by the slope of the shape parameter β of the Weibull-transformed Kaplan–Meier estimates (Figure 4b). This pattern of regularity was lacking in men with TK1 concentrations >0.25 ng/ml: boundaries between age groups did not exist and death started to occur around 2 years after screening irrespective of age (Figure 4c,d). The corresponding figures of PSA show similar behavior although less distinct than that of TK1 (Appendix A).

Survival in the four age groups (stratified by median TK1) was also assessed by the ratio low/high TK1 concentration (Appendix A). The ratio was about 1.0 at the age of 58 years, increased to 1.33 and 1.32 at the age of 62 and 65 years, and decreased to 1.0 again at the age of 68 years. The ratio of the median survival time instead of SP of TK1 was similar, but the SP ratio of PSA was inconsistent (Appendix A).

SP below and above the median TK1 (0.25 ng/ml) is shown in Figure 5. As a reference, the median survival for the four age groups regardless of TK1 and PSA, decreased almost linearly. SP of TK1 decreased with age in convex and concave shapes. Thus, survival became independent from TK1 at the age of 68 years with an SP of about 15 years.

The influence of TK1 content on survival is shown for six concentrations between 0.15 and 0.40 ng/ml (Figure 6). At 0.15 ng/ml the upper limit of 35 years of SP seems to be reached as indicated by plateau values at age 58 and 62 years. Inversely, the SP of 15 years at 0.40 ng/ml and ages of 65 and 68 years indicates the lower limit. Seemingly, this is the age frame in which SP is modified by TK1. The passage from better to worse SP, indicated by the line of the age-related median OS, occurred between 0.21 ng/ml and 0.29 ng/ml. SP at age 62 differed more than 10 years between the TK1 concentration of 0.15 ng/ml and 0.40 ng/ml and then decreased with age. At age 58, SP differed by about 5 years in men with low- and high-TK1 concentration. The TK1 of the four reference groups (0.26, 0.25, 0.25 and 0.24 ng/ml) did not show any association with age.

### 2.4. Survival in Relation to PSA and Age

The relationship between PSA content and SP was estimated in six groups with median concentration of PSA between 1.7 and 11.4 ng/ml (Figure 7). The median (mean) PSA concentration of the four reference groups increased with age from 2.0 (5.0) to 3.4 (7.1), 3.45 (7.4) and 5.7 (12.4) ng/ml. Except for a single data point, the curves representing different PSA levels did not notably deviate from the overall median. Thus, compared with TK1, the level of PSA had a very weak association with survival.

### 2.5. Survival in Relation to TK1 + PSA and Age

To further explore the feasibility of using PSA and TK1 in combination as a prognostic marker, we varied the cut-off value for PSA in the range between 2.0 and 5.0 ng/ml and evaluated the influence of various PSA concentrations on OS in TK1 + PSA group 1 (LowTK1 + LowPSA) and group 4 (HighTK1 + HighPSA). A fixed cut-off concentration for PSA of 3.0 ng/ml was used in further analysis of TK1 + PSA. In screening efforts, this specific cut-off is often used to prompt further clinical evaluation [2,4].

The cohort was further sub-grouped with three alternative cut-off values for TK1: 0.20, 0.25 and 0.30 ng/ml. (Figure 8). The median TK1 concentration of group 1 ranged between 0.17 and 0.20 ng/ml, and that of group 4 ranged between 0.29 and 0.40 ng/ml, whereas the PSA concentration was unchanged (1.8–1.9 and 7.3–7.5 ng/ml, respectively, for group 1 and group 4). The highest difference in SP of more than 10 years was seen in the 62-year-old men with TK1 concentrations of 0.16 and 0.43 ng/ml. In addition, the curves also show a clear separation for the youngest and oldest age groups.

The association between TK1 + PSA and OS was estimated with univariate and multivariate logistic regression analyses (Table 2). Significant predictors for OS were TK1 (*p* = 0.018) and PSA (*p* = 0.001). In addition, age (*p* < 0.0001) and the time between screening and diagnosis were also significant predictors of OS (*p* < 0.0001 and *p* <0.0001). TK1 remained a significant independent variable of OS in the multivariate analysis in the assessment with each of the other variables. In contrast, PSA lost significance in the presence of age (*p* = 0.09) or time between diagnosis of PCa and screening (*p* = 0.27).

We also analyzed all variables separately (Table 3, upper part). Age, time between screening and diagnosis of PCa, PSA and TK1 were all significantly associated with OS. Next, TK1 and PSA was substituted by TK1 + PSA in combination (groups 1–4). These four groups have six different possibilities to interact with each other. There were significant differences between group 4 and group 1 (*p* < 0.0001), 4 and 2 (*p* = 0.016), and group 4 and group 3 (*p* = 0.003) (Table 3, lower part).

In the multivariate analysis, we considered two situations: 1: PCa was diagnosed at the time of PSA and TK1 tests; 2: PCa was detected after PSA and TK1 measurements—a situation such as the one in this study and which enables to evaluate the predictive ability of TK1 + PSA. The first situation requires age only. In the second situation time from blood drawn to diagnosis must be taken into account. Thus, assuming that all PCa were detected simultaneously with the TK1 + PSA measurement (Table 3, multivariate analysis B), the difference in OS between 1 and 4, 2 and 4 and 3 and 4, which was significant in univariate analysis, was also significant in multivariate analysis (*p* = 0.008; *p* = 0.007; *p* = 0.028, respectively).

The difference in OS was, however, lost when the time between screening and diagnosis of PCa was included except for a significant difference between group 2 and 4 (*p* = 0.004) (Table 3, multivariate analysis C). In the Wald test were age (X^2^ = 50.0, *p* < 0.0001), time between screening and diagnosis of PCa (X^2^ = 27.6, *p* < 0.0001) and TK1 + PSA (X^2^ = 8.9, *p* = 0.031), the only significant variables of OS.

### 2.6. Survival of Controls in Relation to TK1 and PSA

As shown in Table 1, the men in the PCa group attained a significantly (*p* = 0.005) higher age than those in the control group. The median values (IQR) at the last follow-up were 84.7 (10.7) and 81.6 (12.5) years. In addition, the number of men who were still alive after 30 years was greater in the PCa group than in the control group. More conspicuously, other malignancies were nearly twice as common (*p* = 0.02) in the control group than in the group with PCa. Furthermore, median time from diagnosis of another malignancy to the last follow-up was shorter (*p* = 0.0007) among the controls (1.0 year) than among the patients who also had PCa (4.9 years).

Table 4 shows the univariate and multivariate analyses of the controls.

In the univariate analysis of the 271 controls, the age at screening (*p* < 0.0001) and the occurrence of malignancy (*p* = 0.02) were significant predictors of OS, whereas TK1 was borderline (*p* = 0.059). However, in the multivariate analysis, all three factors were found to be significant: age (*p* < 0.0001), presence of malignancy (*p* = 0.046) and TK1 (*p* = 0.037).

When excluding controls with other malignancies in a univariate analysis (*n* = 193), age (*p* < 0.0001), TK1 (*p* = 0.015) and PSA (*p* = 0.049) were significant factors for OS. In the multivariate analysis, only age (*p* < 0.0001) and TK1 (*p* = 0.008) were found to be significant predictors of OS, whereas PSA did not reach significance (*p* = 0.14).

For the 78 controls with malignancies, the median (IQR) for TK1 was 0.22 (0.29) ng/ml. This is similar both to the values for the 193 controls without malignancies, 0.24 (0.29) ng/ml, and to that of the 175 men with PCa, which was 0.25 (0.28) ng/ml. Both age at screening and TK1 were found insignificant with the univariate as well as multivariate analysis. The only predictive factor was the time between screening and diagnosis (*p* < 0.0001).

## 3. Discussion

This study focuses on two biomarkers, TK1 and PSA, combined into one TK1 + PSA to predict OS in men with PCa. By a single measurement of TK1 + PSA at screening for PCa, subgroups with 10-year differences in OS could be identified many years prior to diagnosis.

In the combined biomarker, PSA contributes to prostate specificity and its ability to predict future PCa a long time before diagnosis [29,30,31,32,33]. A limitation of PSA is the association with age which was confirmed in this study. The serum level of TK1, on the other hand, was independent of all data at screening including PSA. The predictive capacity of TK1 is unknown at large but we have previously demonstrated an association between TK1 concentration and future prostate cancer-specific mortality [28].

As expected, PSA lost significance as a single parameter in the presence of age but became significant for OS in the combination of TK1 + PSA. The given endpoint of PSA is the diagnosis of PCa. As an endpoint for TK1, OS was chosen. Minor variations in TK1 concentration considerably changed OS. Increased concentration of TK1 identified men at high risk, which suggests the possibility of an individualized risk assessment. The difference in OS with means of TK1 0.15 ng/ml versus 0.40 ng/ml was more than 10 years for men aged 62 years but negligible for men aged 58 and 68 years. However, combinations of cut-off values revealed that using a cut-off value of 3.0 ng/ml for PSA and >0.30 ng/ml for TK1 resulted in good discrimination between high-risk and low-risk cases, with a clear separation of survival probability for the youngest and oldest patients as well. Multivariate analysis revealed no differences between the OS of group 1 (TK1 < 0.25 + PSA < 3.8 ng/ml) and group 2 (TK1 <0.25 + PSA >3.8 ng/ml), and these groups could be combined. Thus, it is possible to categorize most men into two groups.

The relationship between PSA and tumor development is explained by the disruption of the basement membrane and alterations in the basal cell layer of the prostate [34]. Forty years is the calculated time required for the high-grade PINs to reach the palpable size of 1 cm^3^ [35]. The slow growth is the reason that many PCa cases are undetected during life and explains the high frequency of incidental PCa cases found at autopsy or by cystoprostatectomy (about 30% in the age span of interest) [36,37]. The mechanisms of TK1 release into blood circulation are unknown. Against the hypothesis that non-detected in situ carcinomas are the source of increased TK1 concentrations [38] is the finding that the TK1 concentration of the control groups did not differ from that of the group with PCa. Thus, the TK1 concentration during follow-up was unchanged as opposed to PSA with decreasing concentration.

Increased concentrations of TK1 are found in most tumor diseases [19,20] and therefore, TK1 is not specific for PCa. Moreover, low concentrations of TK1 are present in all healthy individuals of all ages, thus, TK1 is neither PCa nor cancer specific [15]. In addition, low-TK1 concentrations can be associated with PCa-related death [28]. Here, we found no difference between TK1 concentrations in the group of men with PCa at screening, those who developed PCa during follow-up, the control group without any malignancies and controls who developed other malignancies (Table 4). Thus, low concentrations of TK1 may be released from sources other than cancer and still affect OS.

We analyzed TK1 in frozen serum stored 30 years after sampling. It is unknown whether the long storage time affected levels of TK1, but it would likely have affected samples from PCa patients and controls equally.

A consequence and an ambition of screening is the early detection of PCa which otherwise would have presented later, presumably with more advanced disease. There is, indeed, a lower frequency of diagnosed PCa in the first years following screening. Therefore, a biological border between screening-detected cases and cases detected during follow-up did not exist. More than 90% of the screening-detected cases were of grade 1 or 2 and bone metastases were detected in only two men [39]. Treatment of the screening-detected cases included radiation therapy, active surveillance, prostatectomy, surgical castration, androgen deprivation or Nd-YAG laser ablation [39]. In this explorative investigation, the variety of treatment and the possible impact of tumor characteristics on PCa diagnosed at screening or during follow-up were not considered but is the subject of a separate investigation. Moreover, the twice as high incidence of malignancies in the control group needs to be evaluated in a larger material.

The aim of large screening trials is to evaluate if PSA-led screening can detect PCa early at an intervenable stage to reduce PCa mortality [2,3,4]. Other PSA-derived markers include the widely used ratio free/total PSA. For instance, at a PSA level < 2.0 ng/ml, free/total PSA was a significant long-term predictor of PCa death [40]. The combination of three forms of PSA: 2proPSA, free/total PSA and total PSA (Beckman Coulter, Health Index) increased the detective power of PSA and free/total PSA [41]. The AUC of the Prostate Health Index, that distinguishes benign from malignant prostate diseases in the PSA range of 2–10 ng/ml, was further enhanced by serum TK1 from 0.73 to 0.78 [25]. The Stockholm 3 model includes, in addition to six plasma protein markers and 232 single nucleotide polymorphisms, clinical and family data. The Stockholm 3 test reduces the number of biopsies needed to identify Gleason >7 PCa by 53% compared with conventional PCa testing [14].

Other markers are less useful or need further evaluation: circulating tumor cells are present in low numbers in non-metastatic PCa [42], or exosomes which are able to transmit miRNAs, DNAs and proteins [43]. Protein biomarkers have also been analyzed in combination with mutations of DNA in the blood circulation [44]. This approach made it possible to identify eight different tumor types at an early stage, although not the most common PCa, with only a few mutations.

## 4. Material and Methods

### 4.1. Cohort and Design

In 1988–1989, from a background population of about 27,000 men living in the catchment area of Stockholm’s South Hospital, 2400 men, aged 55–70 years, were randomly selected to be invited to a PCa screening study. In total, 1782 of these men accepted the invitation, which included evaluation of the prostate by digital rectal (DRE) and ultrasound (TRUS) examinations as well as PSA analysis using the Hybritech Tandem-R method. The screening algorithm stated that abnormal findings on DRE or TRUS and/or PSA levels above 10 ng/ml led to TRUS-guided or randomized quadrant biopsies, and PSA values between 7 and 10 ng/ml prompted reexamination with DRE, TRUS, PSA and biopsy when indicated. Two venous blood samples were drawn: one for PSA analysis, and serum of the other one was drawn after centrifugation and immediately frozen and stored at −80 °C. 1070 samples of the 1782 were randomly selected and the TK1 concentration measured.

For all men, data on PSA, free/total PSA, DRE and prostate size at screening were obtained from local registries; survival time and cause of death were obtained from the Swedish Population-based Registry; and date of the diagnosis of PCa and other malignancies was obtained from the Swedish Cancer Registry. The last follow- up was December 31, 2018. All men in the study signed informed consent to participate and agreed that blood samples were saved for future analysis. The study was approved by the Ethics Review Board (D-no: 2017/1976-32).

PCa was diagnosed at screening or during follow-up in 271 men (15.2%). For each of these, a man of the same age and with a measure of TK1 but otherwise randomly selected constituted the control group. Data on TK1 concentration were available from 175 of the 271 men and the case–control ratio was about 1.5.

PCa diagnosed within 1 year of the screening was defined as screening detected PCa (*n* = 52) and the remaining as follow-up detected PCa (*n* = 123). There was no schedule for follow-up of the participants. Usually, men with symptoms of the prostate were examined by general medical practitioners of the public healthcare system who referred to a specialist in urology if necessary. Autopsies were uncommon but 4 autopsy-detected cases were included in which PCa was the reason of death.

The median values of TK1 and PSA serum concentrations measured once in two blood samples about 25 years apart from each other were used to stratify the men into two groups each. TK1 was combined then with PSA into four groups: TK1 < 0.25 ng/ml + PSA < 3.8 ng/ml, TK1 < 0.25 ng/ml + PSA > 3.8 ng/ml, TK1 > 0.25 ng/ml + PSA < 3.8 ng/ml and TK1 > 0.25 ng/ml + PSA > 3.8 ng/ml. Other cut-off values for TK1 and PSA were also considered. To investigate the significance of age, the men were stratified into 4 groups by quartiles of age at screening.

### 4.2. Determination of TK1 Concentration

All samples were transported from their storage at the hospital bio bank to the Department of Anatomy, Physiology and Biochemistry, at the Swedish University of Agricultural Sciences, Uppsala, Sweden. The samples were thawed on arrival at the laboratory, aliquoted into 200 µL and restored in the department bio bank until analysis with the enzyme-linked immunoassay (ELISA) TK 210 of AroCell AB, Uppsala. This test is based on two monoclonal antibodies against the C-terminal region of TK1 and was performed in duplicates according to the manufacturer‘s instructions (www.e-labeling.eu/ARO1001--15-7, 5 December 2019). Briefly, a specific antibody binds to the epitope of TK1, and a second specific antibody binds to the target molecule which initiates a light emission reaction. The samples were analyzed together with two internal controls and a quality control. The overall coefficient of variation in the controls and quality controls between kits was in the range of 3.5–9.6%. All samples were randomly selected and blinded to identity primary data and outcomes.

### 4.3. Statistical Analysis

Differences in baseline data and results between subgroups were examined with Fisher‘s exact test and the Wilcoxon test, and associations between variables were evaluated by regression analysis. OS was assessed by the Kaplan–Meier method, and differences in survival were determined with the log-rank and Wilcoxon tests. Because the number of men in the subgroups was partly limited or median survival time was not reached, the survival probability (SP) was estimated from Weibull-transformed Kaplan–Meier estimates. The scale factor α of these Weibull fits was well correlated with the medium survival time (Appendix A). Logistic regression analysis was used to evaluate the significance of various parameters for OS. For calculations of SP and statistical analyses, the software of JMP15.1.0, SAS, Cary, NC, USA was used. A *p*-value ≤ 0.05 indicated statistical significance.

## 5. Conclusions

Serum concentrations of TK1 and PSA can both predict long-term risk in PCa. The combined biomarker TK1 + PSA increases the possibility of identifying subgroups with different OS. This may allow the individualized treatment of patients. Since OS can be predicted a long time before PCa tumors are visible, the combined biomarker TK1 + PSA may reveal an unknown disposition of OS and is a new dimension for further studies of TK1.

## Figures and Tables

**Figure 1 ijms-24-05160-f001:**
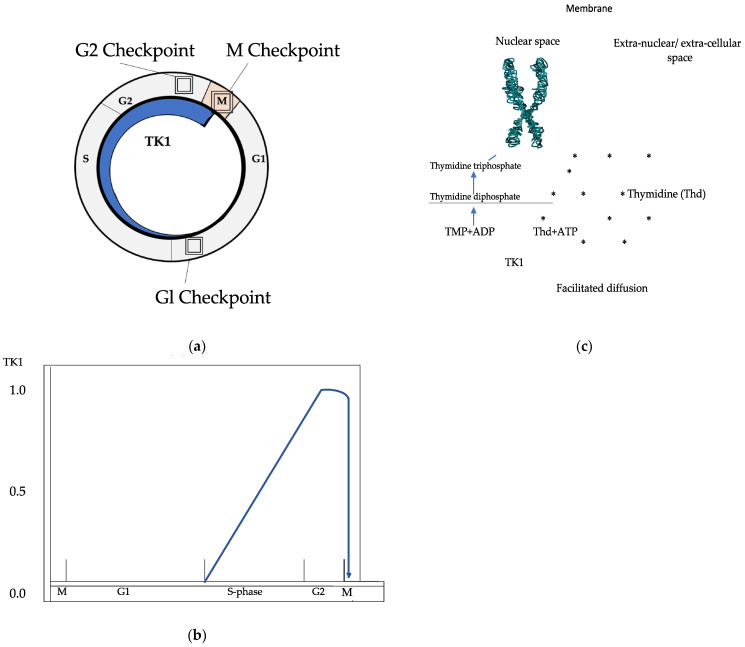
Overview of the function of thymidine kinase 1 in the DNA synthesis. (**a**) TK1 and cell cycle, (**b**) TK1 concentration during cell cycle, (**c**) transformation of thymidine* (Thd) to deoxythymidine—monophosphate.

**Figure 2 ijms-24-05160-f002:**
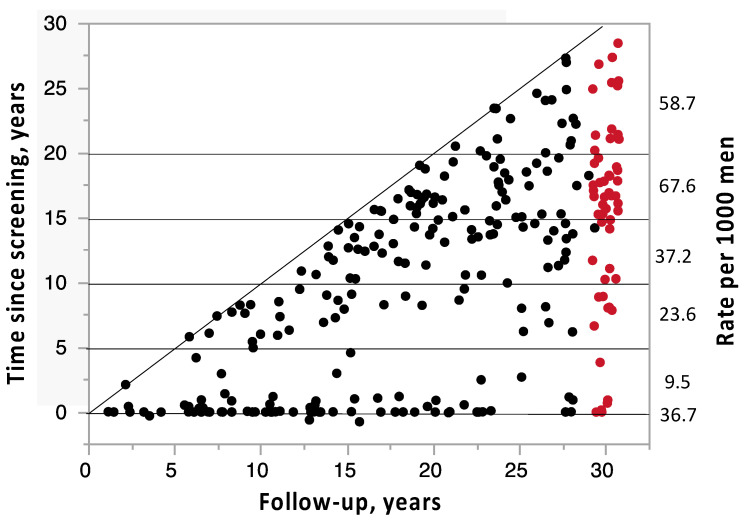
Last follow-up in relation to the time of the diagnosis in 271 men with prostate cancer, detected by screening 1988/89 and during 30 years of follow-up. Red symbols: alive, black symbols: deceased. On the right side, number of prostate cancers in relation to the actual number of the initial 1782 men.

**Figure 3 ijms-24-05160-f003:**
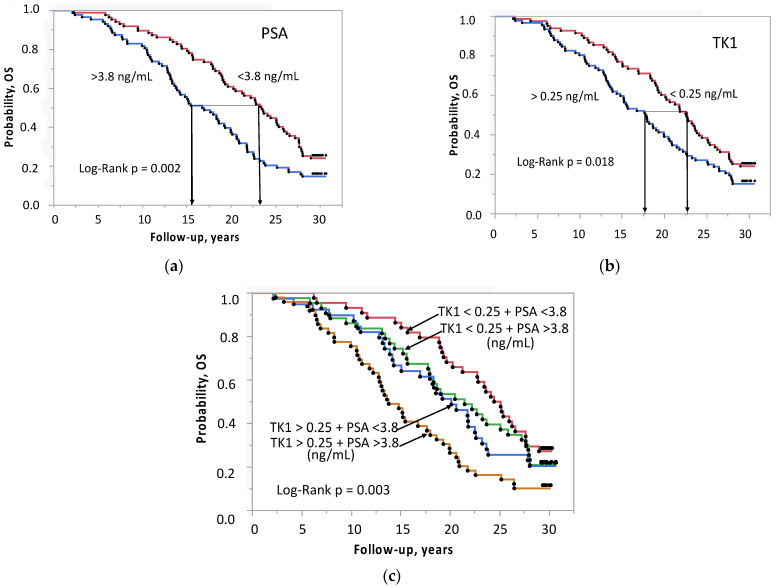
Overall survival by Kaplan–Meier estimates in 175 men with prostate cancer, (**a**) according to the median concentration of PSA (**b**), according to the median concentration of TK1, (**c**) according to TK1 + PSA. 1. TK1 < 0.25 + PSA < 3.8 ng/ml, 2. TK1 < 0.25 + PSA > 3.8 ng/ml, 3. TK1 > 0.25 + PSA < 3.8 ng/ml, 4. TK1 > 0.25 + PSA > 3.8 ng/ml. Abbreviations: PSA = prostate-specific antigen, TK1 = thymidine kinase 1.

**Figure 4 ijms-24-05160-f004:**
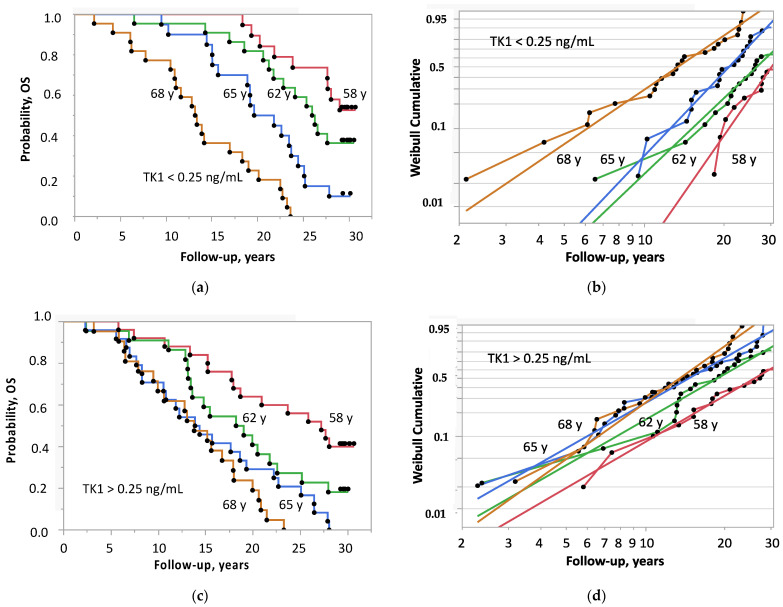
Overall survival of 175 men with prostate cancer stratified by quartiles of age at screening and median TK1 concentration: (**a**) Kaplan–Meier estimates of TK1 concentrations < 0.25 ng/ml; (**b**) Weibull-transformed Kaplan–Meier estimates of TK1 concentrations < 0.25 ng/ml; (**c**) Kaplan–Meier estimates of TK1 concentrations > 0.25 ng/ml; (**d**) Weibull-transformed Kaplan–Meier estimates of TK1 concentrations > 0.25 ng/ml.

**Figure 5 ijms-24-05160-f005:**
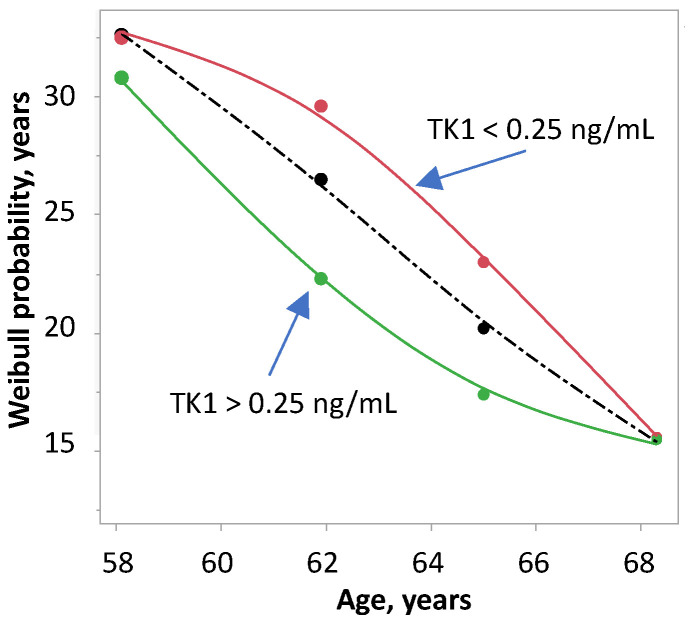
Survival probability of 175 men with prostate cancer stratified by age and TK1 below and above the median. The survival probability was calculated from Weibull-transformed Kaplan–Meier estimates. The shattered line shows the survival probability of age alone.

**Figure 6 ijms-24-05160-f006:**
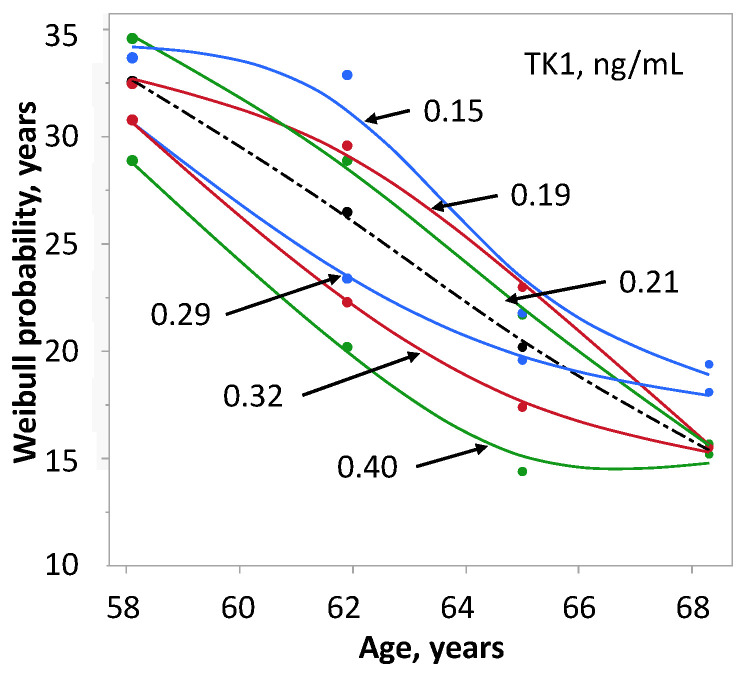
Influence of TK1 concentration on survival probability of 175 men with prostate cancer stratified by age. The survival probability was calculated from Weibull-transformed Kaplan–Meier estimates. The shattered line shows the survival probability of age alone.

**Figure 7 ijms-24-05160-f007:**
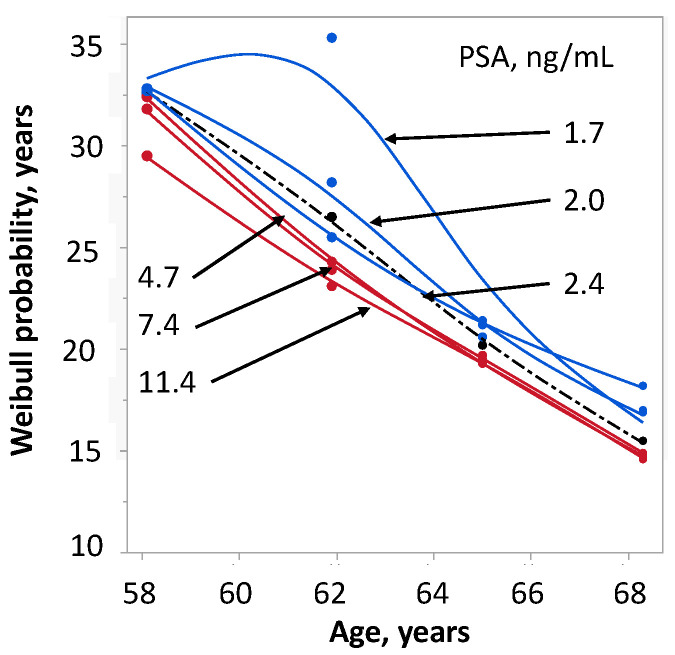
Influence of PSA concentration on survival probability of 175 men with prostate cancer stratified by age. The survival probability was calculated from Weibull-transformed Kaplan–Meier estimates. The shattered line shows the survival probability of age alone.

**Figure 8 ijms-24-05160-f008:**
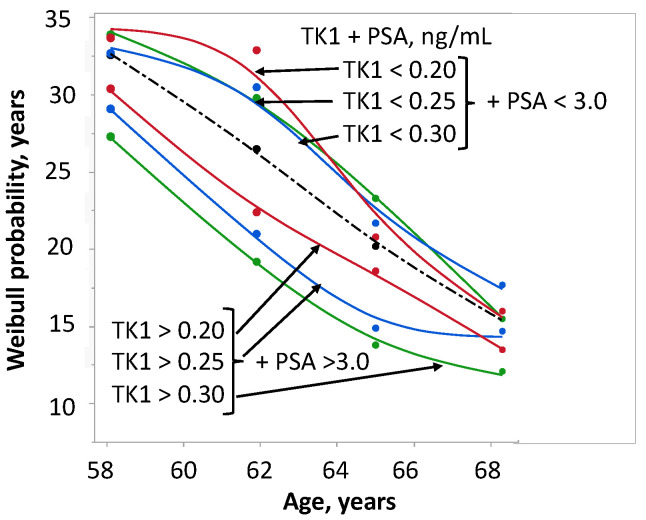
Survival probability of 175 men with prostate cancer stratified by age and different cut-off values of TK1 concentration in combination with PSA below and above 3.0 ng/ml.

**Table 1 ijms-24-05160-t001:** Characteristics of 271 men with prostate cancer and 271 controls. Median values (IQR). Prostate-related characteristics and TK1 were obtained in connection with screening 1988–1989. Abbreviations: PSA = prostate-specific antigen, TK1 = thymidine kinase 1.

Variable	Prostate Cancer	Controls	*p*
Age at screening	63.2 (6.8)	63.7 (6.5)	0.172
Age last follow-up	84.7 (10.7)	81.6 (12.5)	**0.0005**
Alive last follow-up, n	58 (20.7%)	36 (13.2%)	**0.022**
PSA, ng/ml	3.2 (4.5)	1.8 (0.11)	**<0.0001**
Free/total PSA	0.13 (0.09)	0.15 (0.08)	**0.0015**
Prostate volume, cm^3^	23.3 (12.0)	21.4 (10.4)	**0.004**
PSA density, ng/ml/cm^3^	0.09 (0.05)	0.14 (0.15)	**<0.0001**
TK1, ng/ml *n* = 175	0.25 (0.14)	0.23 (0.16)	0.607
Other malignancies, *n* 48 (17.7%)	78 (28.8%)		**0.002**
Age	77.6 (9.3)	76.5 (11.4)	0.843
Time to diagnosis, years	14.7 (11.4)	13.6 (10.0)	0.284
Time between diagnosis and last follow-up, years	4.89 (1.9)	1.02 (4.8)	**0.0007**

**Table 2 ijms-24-05160-t002:** Association between TK1 and PSA, age and time between screening and diagnosis of prostate cancer.

Variable	OR 95% CI	X^2^	*p*
**Univariate analysis**			
Age at screening, per year	1.23 (1.17–1.29)	61.4	**<0.0001**
Time screening—Diagnosis, per year	0.93 (0.91–0.95)	41.3	**<0.0001**
PSA <3.0>, ng/ml	1.74 (1.24–2.45)	10.3	**0.0001**
TK1 <0.25>, ng/ml	1.49 (1.07–2.08)	5.5	**0.018**
**Multivariate analysis**			
TK1 <0.25>, ng/ml	1.46 (1.04–2.03)	4.9	**0.026**
+Age at screening, per year	1.23 (1.17–1.29)	61.3	**<0.0001**
TK1 <0.25>, ng/ml	1.55 (1.11–2.17)	6.6	**0.010**
+Time screening—Diagnosis, per year	0.93 (0.91–0.95)	42.3	**<0.0001**
TK1 <0.25>, ng/ml	1.53 (1.10–2.13)	6.2	**0.012**
+PSA <3.0>, ng/ml	1.78 (0.26–2.50)	11.0	**0.0009**
PSA <3.0>, ng/ml	1.35 (0.95–1.91)	2.9	0.089
+Age at screening, per year	1.22 (1.16–1.29)	59.9	**<0.0001**
PSA <3.0>, ng/ml	1.22 (0.85–1.75)	1.2	0.27
+Time screening—Diagnosis, per year	0.15 (0.08–0.29)	32.3	**<0.0001**
Age at screening, per year	1.20 (1.14–1.27)	49.2	**<0.0001**
Time screening—Diagnosis, per year	0.94 (0.92–0.96)	28.3	**<0.0001**

**Table 3 ijms-24-05160-t003:** Evaluation of baseline data by univariate and multivariate analyses for overall survival in 175 men with prostate cancer. *The biomarker TK1+PSA has 4 subgroups.

Variable	Univariate Analysis		Multivariate Analysis A		Multivariate Analysis B		Multivariate Analysis C	
	OR 95% CI	*p*	OR 95% CI	*p*	OR 95% CI	*p*	OR 95% CI	*p*
Age, per year	1.23 (1.17–1.29)	**<0.0001**	1.21 (1.15–1.28)	**<0.0001**	1.23 (0.17–1.30)	**<0.0001**	1.22 (1.15–1.78)	**<0.0001**
Time screening- diagnosis of PCa, per year	0.930 (91–0.95)	**<0.0001**	0.94 (0.91–0.96)	**<0.0001**			0.94 (0.91–0.96)	**<0.0001**
PSA > 3.0 <, ng/ml	1.74 (1.24–2.45)	**0.001**	1.07 (0.69–1.65)	0.77				
Free/total PSA < 0.13 >	0.95 (0.68–1.33)	0.77	1.22 (0.82–1.81)	0.34	1.11 (0.75–1.64)	0.59	1.25 (0.84–1.86)	0.27
Prostate volume, > 0.23<, cm^3^	1.15 (0.83–1.61)	0.40	1.16 (0.80–1.70)	0.44	1.02 (0.70–1.49)	0.92	1.15 (0.79–1.67)	0.48
Other malignancies, yes/no	0.84 (0.54–1.31)	0.45	1.10 (0.70–1.74)	0.68	1.14 (0.72–1.81)	0.57	1.09 (0.69–1.73)	1.14
TK1 > 0.25 < ng/ml	1.49 (1.07–2.08)	**0.018**	1.61 (1.13–2.28)	**0.008**				
*TK1 <0.25 > + PSA <3.0 >, ng/ml	Sub-group 1: TK1 < 0.25 + PSA < 3.0 ng/ml		Sub-group 2: TK1 < 0.25 + PSA > 3.0 ng/ml		Sub-group 3: TK1 > 0.25 + PSA < 3.0 ng/ml		Sub-group 4: TK1 > 0.25 + PSA > 3.0 ng/ml	

4 vs. 1	2.61 (1.61–4.23)	**<0.0001**			2.09 (1.21–3.69)	**0.008**	1.62 (0.92–2.86)	0.09
4 vs. 2	1.69 (1.10–2.60)	**0.016**			1.85 (0.18–2.89)	**0.007**	1.94 (1.24–3.02)	**0.004**
4 vs. 3	1.99 1.26–3.16)	**0.003**			1.82 (1.07–3.10)	**0.028**	1.35 (0.78–2.34)	0.29
2 vs. 1	1.54 (0.93–2.55)	0.09			1.13 (0.66–1.93)	0.65	0.84 (0.48–1.45)	0.53
3 vs. 1	1.31 (0.77–2.22)	0.31			1.15 (0.67–1.97)	0.61	1.20 (0.70–2.07)	0.50
3 vs. 2	0.85 (0.52–1.38)	0.51			1.01 (0.59–1.73)	0.95	1.44 (0.83–2.50)	0.20

Multivariate **A**: baseline data, exclusive TK1 + PSA; multivariate **B**: TK1 <0.25> + <PSA> 3.0, ng/ml exclusive time screening diagnosis of PCa; multivariate **C**: TK1 <0.25> + <PSA> 3.0, ng/ml inclusive time screening diagnosis of PCa. Abbreviations: TK1 = thymidine kinase 1, PSA = prostate-specific antigen, OR = odds ratio, PCa = prostate cancer.

**Table 4 ijms-24-05160-t004:** Evaluation of baseline data by univariate and multivariate analysis for overall survival in 271 controls without prostate cancer.

Variable		Univariate Analysis		Multivariate Analysis	
		OR 95% CI	*p*	OR 95% CI	*p*
**All controls**	Age, per year	1.07 (1.04–1.10)	**<0.0001**	1.07 (1.03–1.10)	**<0.0001**
(*n* = 271)	Malignancies, yes/no	1.42 (1.07–1.87)	**0.02**	1.32 (1.0–1.76)	**0.046**
	TK1, ng/ml per unit	1.51 (0.92–2.19)	0.059	1.50 (0.97–2.28)	**0.037**
	PSA, ng/ml per unit	1.20 (0.87–1.56)	0.231	1.02 (0.98–1.04)	0.23
**Controls without malignancies**					
(*n* = 193)	Age, per year	1.08 (1.04–1.12)	**<0.0001**	1.08 (1.04–1.12)	**<0.0001**
	TK1, ng/ml per unit	1.66 (1.03–2.37)	**0.015**	1.73 (1.08–2.46)	**0.008**
	PSA, ng/ml per unit	1.03 (0.99–1.05)	**0.049**	1.02 (0.98–1.05)	0.136
Prostate volume,					
cm^3^ per unit		1.01 (0.99–1.02)	0.563	0.99 (0.98–1.01)	0.69
**Controls with malignancies**					
(*n* = 78)	Age, per year	1.04 (0.97–1.09)	0.30	1.02 (0.96–1.08)	0.60
	TK1, ng/ml per unit	0.46 (0.07–2.19)	0.39	0.38 (0.05–1.77)	0.27
	Diagnose, time since				
	screening, per year.	0.91 (0.88–0.94)	**<0.0001**	0.91 (0.88–0.94)	**<0.0001**
	PSA, ng/ml per unit	0.96 (0.87–1.04)	0.31	0.99 (0.89–1.08)	0.82

## Data Availability

The data generated during the current study are available from the corresponding authors upon reasonable request.

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
