# Peer review of "Prediction of Overall Survival by Thymidine Kinase 1 Combined with Prostate-Specific Antigen in Men with Prostate Cancer"

_ijms, 2023, doi:10.3390/ijms24065160_

Round 1

Reviewer 1 Report

With great interest, I read through this prospective study on TK1 and PSA over more than 20 years of median follow-up. However, the authors didn't convey a clear message for the readers, although they have compelling data. I would strongly suggest the authors could consider re-organizing the structure of the article. 

Another main point is if the authors can provide additional information regarding prostate cancer, for example, Gleason score, so far one of the most important prognostic parameters. 

Author Response

We are grateful for a thorough scrutiny of our manuscript. We have found the reviewer’s comments pertinent and helpful and have revised the manuscript accordingly.  

The reviewer askes for a clear message to the reader and we have tried to convey the two main messages: 1. TK1 is obviously a strong parameter for future outcome in prostate cancer which can improve the predictive value of PSA. 2. TK1 is however, nonspecific for prostate cancer. This insight gives a new dimension for further studies of TK1.

We agree that Gleason score and other tumor characteristics are important parameters for outcome of prostate cancer disease. The problem in this study is that the techniques of biopsy, definitions of malignancies, as well as treatment, have changed profoundly during the more than 30 years period of the investigation. This is clearly mentioned in the discussion. 

Reviewer 2 Report

The manuscript entitled (Prediction of overall survival by thymidine kinase 1 combined with prostate-specific antigen in men with prostate cancer) submitted by Tribukait et al. shows that low TK1 in the blood circulation indicates the release of TK1 from sources other than cancers, but significance for OS. The authors did a prediction of overall survival by thymidine kinase 1 combined with prostate-specific antigen in men with prostate cancer. The overall study is well-presented and written. However, some improvements need to be addressed to be accepted in IJMS.

1. The English editing of the manuscript is highly recommended.

2. A figure in the introduction section illustrates a brief background of thymidine kinase signaling.

Author Response

We are grateful for a thorough scrutiny of our manuscript. We have found the reviewer’s comments pertinent and helpful and have revised the manuscript accordingly.

  1. We hope that the improvement of the English language has also clarified the message in the manuscript.
  2. We have added a figure in the introduction section of the role of TK1 in DNA-synthesis may provide further background to the study.
  3. I agree with the referee that the study design has short-comes. A main limitation was the restricted number of prostate cancers. Thus, the results should be confirmed in larger materials.

Round 2

Reviewer 1 Report

No further comments. 

Reviewer 2 Report

The authors have addressed my comments. Please add the supplementary figures and Tables in a separate supplementary material file. The article can be accepted.